# Vascular Wall–Mesenchymal Stem Cells Differentiation on 3D Biodegradable Highly Porous CaSi-DCPD Doped Poly (α-hydroxy) Acids Scaffolds for Bone Regeneration

**DOI:** 10.3390/nano10020243

**Published:** 2020-01-29

**Authors:** Monica Forni, Chiara Bernardini, Fausto Zamparini, Augusta Zannoni, Roberta Salaroli, Domenico Ventrella, Greta Parchi, Micaela Degli Esposti, Antonella Polimeni, Paola Fabbri, Fabio Fava, Carlo Prati, Maria Giovanna Gandolfi

**Affiliations:** 1Department of Veterinary Medical Sciences, University of Bologna, Ozzano Emilia, 40064 Bologna, Italy; monica.forni@unibo.it (M.F.); chiara.bernardini5@unibo.it (C.B.); augusta.zannoni@unibo.it (A.Z.); roberta.salaroli@unibo.it (R.S.); domenico.ventrella2@unibo.it (D.V.); 2Laboratory of Biomaterials, Green Materials and Oral Pathology, School of Dentistry, Department of Biomedical and Neuromotor Sciences, University of Bologna, 40125 Bologna, Italy; fausto.zamparini2@unibo.it (F.Z.); greta.parchi@studio.unibo.it (G.P.); 3Department of Civil, Chemical, Environmental and Materials Engineering, University of Bologna, 40136 Bologna, Italy; micaela.degliesposti@unibo.it (M.D.E.); p.fabbri@unibo.it (P.F.); fabio.fava@unibo.it (F.F.); 4Department of Oral and Maxillo-facial Sciences, Pediatric Dentistry Unit, Sapienza University of Rome, 00161 Rome, Italy; antonella.polimeni@uniroma1.it; 5Endodontic Clinical Section, School of Dentistry, Department of Biomedical and Neuromotor Sciences, University of Bologna, 40125 Bologna, Italy; carlo.prati@unibo.it

**Keywords:** vascular wall–mesenchymal stem cells, biodegradable mineral scaffolds, engineered tissue, angiogenesis, oral bone defects, polylactic acid (PLA), poly-e-caprolactone (PCL), green biomaterials, green scaffolds, biobased materials

## Abstract

Vascularization is a crucial factor when approaching any engineered tissue. Vascular wall–mesenchymal stem cells are an excellent in vitro model to study vascular remodeling due to their strong angiogenic attitude. This study aimed to demonstrate the angiogenic potential of experimental highly porous scaffolds based on polylactic acid (PLA) or poly-e-caprolactone (PCL) doped with calcium silicates (CaSi) and dicalcium phosphate dihydrate (DCPD), namely PLA-10CaSi-10DCPD and PCL-10CaSi-10DCPD, designed for the regeneration of bone defects. Vascular wall–mesenchymal stem cells (VW-MSCs) derived from pig thoracic aorta were seeded on the scaffolds and the expression of angiogenic markers, i.e. CD90 (mesenchymal stem/stromal cell surface marker), pericyte genes α-SMA (alpha smooth muscle actin), PDGFR-β (platelet-derived growth factor receptor-β), and NG2 (neuron-glial antigen 2) was evaluated. Pure PLA and pure PCL scaffolds and cell culture plastic were used as controls (3D in vitro model vs. 2D in vitro model). The results clearly demonstrated that the vascular wall mesenchymal cells colonized the scaffolds and were metabolically active. Cells, grown in these 3D systems, showed the typical gene expression profile they have in control 2D culture, although with some main quantitative differences. DNA staining and immunofluorescence assay for alpha-tubulin confirmed a cellular presence on both scaffolds. However, VW-MSCs cultured on PLA-10CaSi-10DCPD showed an individual cells growth, whilst on PCL-10CaSi-10DCPD scaffolds VW-MSCs grew in spherical clusters. In conclusion, vascular wall mesenchymal stem cells demonstrated the ability to colonize PLA and PCL scaffolds doped with CaSi-DCPD for new vessels formation and a potential for tissue regeneration.

## 1. Introduction

Vascularization is a crucial factor when approaching any engineered tissue [1]. A proper vascular network should allow the transport of oxygen, nutrients and growth factors, as the diffusion limit to maintain viable cells is only 400 μm [2]. Moreover, vascular network should also act as a transport system for hormones, waste products, toxic and non-functional substances [3], and should be a source of growth factors locally produced by endothelial cells, such as pro-angiogenic morphogenetic factors [4].

In post-natal life, vascular remodeling is main due angiogenesis process. This process takes place through cell recruitment and cell differentiation [4], with the involvement of growth factors such as vascular endothelial growth factor (VEGF) and fibroblast growth factor-2 (FGF-2) [5].

Many tissue-specific stem cells/progenitors could be involved in neoangiogenesis [6], as vascular stem cells and vascular wall-mesenchymal stem cells, residing in the wall of the large vessels, possess excellent characteristics in inducing new vascularization [7,8,9] and are a promising resource in regenerative medicine including tissue engineering. 

A new method to isolate and culture vascular wall mesenchymal stem cells (VW-MSCs) from the tunica media of pig thoracic aorta was previously described [10]. These cells have excellent pro-angiogenic features either for their ability of differentiating in endothelial cells or capacity to sustain a capillary network [11]. Moreover, the pro-angiogenic features of VW-MSCs secretome have been demonstrated [12]. Overall, VW-MSCs are an excellent model for translational medicine for the high similarity between human and pig.

Biodegradable synthetic polymers such as polylactic acid (PLA) and poly-e-caprolactone (PCL) are currently used for scaffold design [13,14,15]. 

PLA is a green bioplastic produced through the bacteria fermentation of polysaccharide-based natural resources (such as corn or starch), with the final result of lactic acid (monomer) and/or lactide (dimer) polymerization [16]. PLA is used for different biomedical applications (including orthopedic screws, fracture fixation devices and drug delivery applications) [13,15].

PCL is a biodegradable polyester, approved as implantable biomaterial by FDA and for different applications in human body [14]. PCL is used in medicine for its properties of biocompatibility, biodegradation and collagen stimulator [17]. 

Combination of biodegradable synthetic polymers with inorganic materials resulted in hybrid biomaterials with suitable properties for tissue engineering [13,14].

Interestingly it has been reported that some ions have a significant role in neo-angiogenesis. Silicon (Si) is an important element able to induce angiogenesis during new bone formation and to accelerate bone regeneration [5,18].

Bioactive calcium silicate-based materials (CaSi) may provide interesting advantages in this context in relationship to their chemistry as they expose silanol groups and release silicon [18,19,20]. CaSi demonstrated biointeractive properties [21,22], ion release (Ca and OH) [23] and the ability to induce the differentiation of different populations of cells, such as orofacial bone mesenchymal stem cells [24], cementoblasts [25], pulp cells [26], mesenchymal stem cells [27], and oral derived periapical cyst mesenchymal stem cells [28]. 

The inclusion of calcium phosphates, such as dicalcium phosphate dihydrate (DCPD) into CaSi materials demonstrated to enhance their biological properties and apatite-forming ability [21,22,24].

Moreover, when CaSi are used as filler in a polymeric matrix, such as PLA or PCL, the high alkalizing ability may counterbalance the acidic degradation products of synthetic poly-α-hydroxyl polymers [29,30]. These properties support their role as filler in tissue engineering. Some scaffolds have been designed for angiogenic and osteogenic purposes containing bioactive calcium silicates [31]. Recently, highly porous PLA and PCL scaffolds doped with bioactive calcium silicates and calcium phosphates have been designed [28,29,30]. Mineral-doped PLA scaffolds demonstrated the ability to be colonized by oral derived mesenchymal stem cells and to stimulate their shift through osteogenic lineage [28]. Highly porous mineral-doped PCL scaffolds showed interesting biointeractive properties and the ability to nucleate apatite, enhancing the biological properties of pure PCL [30].

The aim of the present study was to evaluate the ability of VW-MSCs to colonize experimental highly porous based mineral-doped poly-α-hydroxyl scaffolds, and to express their angiogenic potential. PLA- and PCL-based scaffolds doped with CaSi and DCPD have been used as 3D model and tested for the gene expression of angiogenic markers CD 90, α-SMA, PDGFR-β and NG2 of VW-MSCs. Pure PLA and PCL scaffolds and cell plastic were used as control (3D in vitro model vs. 2D in vitro model).

## 2. Materials and Methods 

### 2.1. Scaffolds Preparation 

PLA (IngeoTM biopolymer PLA 4060D, Natureworks LLC, Blair, NE, USA) having average Mw = 65,000 g/mol was received in pellet form. PCL (Sigma-Aldrich, Milan, Italy) having average Mw = 45,000 g/mol was received in pellet form.

Prior their use, the synthetic polymers were purified by means of dissolution in 10% wt/vol (for PLA) or 15% wt/vol (for PCL) of CHCl_3_ solutions (HPLC grade, Sigma-Aldrich, Milan, Italy), followed by reprecipitation in pre-cooled methanol (Sigma-Aldrich, Milan, Italy) to eliminate residual polymerization catalysts. 

CaSi, composed of dicalcium and tricalcium silicates, tricalcium aluminate and calcium sulfate (Aalborg, Denmark) and DCPD (CaHPO_4_·2H_2_O, Sigma-Aldrich, Steinheim, Germany) powders were obtained by melt-quenching technique and milling procedures [26] and used as mineral fillers.

The scaffolds were prepared through TIPS according to a previous protocol [28,29,30]. PLA or PCL solutions in 1,4-dioxane were obtained, reaching a 3.5% wt/vol concentration. CaSi and DCPD mineral powders were added in 10% by weight with respect to PCL or PLA. The solutions were sonicated for 3 h using a ultrasonic processor UP50H (Hielsher, Teltow, Germany; operative parameters 50 W, 30 kHz) with a 2 mm sonotrode MSD titanium tip, then put inside 60 mm plates and cooled at −18 °C for 18 h. Finally, the frozen samples were extracted and immersed in an ethanol bath (Sigma-Aldrich, Milan, Italy) precooled at −18 °C for 18 h, with a solvent refresh every 3 h [29,30].

For each formulation, 6 disks (diameter 60 ± 1 mm, thickness 10 ± 0.1 mm) were prepared, namely PLA, PLA-10CaSi-10DCPD, PCL, and PCL-10CaSi-10DCPD.

### 2.2. Surface Porosity Evaluated by the Morphometric Analysis on ESEM Images

The scaffolds were cut into samples (10.00 ± 1.00 mm height, 15.00 ± 1.00 mm length and 10.00 ± 1.00 mm width; n = 3 per formulation) and examined using an environmental scanning electron microscope (ESEM, Zeiss EVO 50; Carl Zeiss, Oberkochen, Germany). The samples were placed directly onto the ESEM stub without any preparation (uncoated) and investigated in wet conditions at low vacuum (100 Pa) in Quadrant Back-Scattering Detector (QBSD) (operative parameters: 20 kV accelerating voltage, 8.5 mm working distance, 0.5 wt% detection level, 133 eV resolution, 100 μs amplification time and 60 s measuring time).

The images were analyzed using Image J program (National Institutes of Health, Bethesda, MY, USA) to evaluate the surface porosity of the scaffolds, calculated as the ratio between the blackest areas (corresponding to the micropores) and the total investigated area [29,32].

Measurements were made in triplicate on three different areas at both 500× and 1000× magnification, and the mean value for each magnification was recorded.

### 2.3. Isolation and Culture of Primary Porcine Vascular Wall Mesenchymal Stem Cells

Three-month old female pigs (Large White) were euthanized for different experimental investigation and primary VW-MSCs were isolated from the thoracic aorta in order to generate three primary cell culture replicates following a previously described methodology [11,33].

All procedures on animals were reviewed and approved in advance by the Ethical Committee of the University of Bologna (Bologna, Italy) and then approved by the Italian Ministry of Health (Protocol number n.43-IX/9 all.37; 20/11/2012).

Plastics used for standard culture condition were from BD-Falcon (Corning, NY, USA). DAPI solution for DNA staining and antibiotic–antimycotic Dulbecco’s phosphate buffered saline (DPBS, Gibco-Life Technologies, Carlsbad, CA, USA).

Trypsin–EDTA solution 1X was purchased from Sigma-Aldrich (St. Louis, MO, USA). Growth medium (Promocell Pericyte Growth Medium) (PGM, Promocell GmbH, Heidelberg, Germany).

All the experiments were performed with cells at passage 3 and cultured in PGM (Heidelberg, Germany).

### 2.4. Cell Seeding Efficiency Assay

All the scaffolds were sterilized by 30 min of incubation in absolute EtOH followed by three washes in DPBS (30 min each one). Then cubic-shaped sections of about 1 mm on each side were produced using scalpel blades 30 sections were inserted into a microtube containing 500 μL of PGM culture medium and incubated over night at 4 °C. The day after, VW-MSCs were seeded on different biomaterials (PLA-10CaSi-10DCPD and PCL-10CaSi-10DCPD) following a published method [34] with some main adaptations. Microtubes with the biomaterials were incubated to reach a temperature of 38 °C. The medium was then separated, and biomaterials were drop-seeded with 100 µL of a concentrated cell suspension containing 4 × 10^5^ or 8 × 10^5^ cells.

The microcentrifuge tubes were placed on a rocker oscillating at 30 rpm for 2 h to permit initial cell attachment then 1 mL of fresh cell culture medium was slowly added to each tube and cells were cultured for additional 24, 48, 72 h in static condition at 38.5 °C at 5% CO_2_. Cell-free biomaterials were incubated under same conditions and used as control.

Cell seeding efficiency (CSE) was estimated using an indirect method [34] after 24, 48 and 72 h. Briefly, CSE was calculated using the following equation: CSE (%) = (1 − cells_u_/cells_i_) × 100, where cells_i_ is the number of cells initially seeded and cells_u_ is the number of unattached cells in the residual medium and in DPBS used for rinsing cell-seeded biomaterials. Unattached cells were counted by hemocytometer in three different aliquots of medium collected from each sample.

### 2.5. Metabolic Cell Activity Assay

The metabolic activity of VW-MSCs was monitored at 24, 48 and 72 h using MTT based assay (Sigma-Aldrich, MO, USA), following the manufacturer’s instruction, with some main adaptation. After the removal of the medium, scaffolds were washed two times with DPBS. Then, 20 µL of MTT substrate was added and recovered for 4 h at 38.5 °C at 5% CO_2_. After that, 200 µL of solubilisation solution was added and, after 30 min, the solution was vigorously mixed. Finally, the absorbance was measured spectrophotometrically at a wavelength of 570 nm, with the background subtraction at 690 nm.

### 2.6. RNA Extraction and qPCR

TRI Reagent (Molecular Research Center, Inc, Cincinnati, OH, USA) and NucleoSpin RNA II kit (Macherey-Nagel GmbH & Co. KG, Düren, Germany) were purchased and used for RNA extraction.

Briefly, culture medium was removed after 72 h and different scaffolds where washed twice with DPBS; 500 mL of TRI Reagent (Molecular Research Center, Inc.) was supplemented and the materials were lysed using a TH Tissue Homogenizer (Omni International, GA, USA).

Subsequently, 100 μL of chloroform was added to the solution, mixed and incubated at room temperature for 10 min. Samples were then centrifuged at 12,000× *g* for 10 min to recover the aqueous phase.

Absolute ethanol (99%) was added in equal volume to the solution and then submitted to the Nucleo spin RNA Column. RNA was finally purified according to the manufacturer’s instructions.

Total RNA (500 ng) was quantified spectrophotometrically (Denovix, Denovix Inc., Wilmington, NC, USA) and then reverse-transcripted to cDNA using the iScript cDNA Synthesis Kit (Bio-Rad Laboratories Inc., Hercules, CA, USA) in a final volume of 20 μL.

Selection of swine primers was performed using Beacon Designer 2.07 (Premier Biosoft International, Palo Alto, CA, USA).

Quantitative real-time PCR (qPCR) was carried out to analyze gene expression profile in CFX96 thermal cycler (Bio-Rad, Hercules, CA, USA), using a multiplex real time reaction for reference genes (namely glyceraldehyde-3-phosphate dehydrogenase, hypoxanthine guanine phosphoribosyl transferase and β-Actin). Taq-Man probes and SYBR green detection were used for the expression of the following target pericyte genes: CD90 (mesenchymal stem/stromal cell surface marker), α-SMA (alpha smooth muscle actin), PDGFR-β (platelet-derived growth factor receptor-β), and NG2 (neuron-glial antigen 2).

All amplification reactions were performed in 20 μL and analyzed in duplicates (10 μL/well). Multiplex PCR contained: 10 μL of iTaqMan Probes Supermix (Bio-RAD), 1 μL of forward and reverse primers (5 μM each) of each reference gene, 0.8 μL of iTaq-Man Probes (5 μM) of each reference gene, 2 μL cDNA and 2.6 μL of water. The following temperature profile was used: initial denaturation at 95 °C for 30 s followed by 40 cycles of 95 °C for 5 s and 60 °C for 30 s.

The SYBR Green reaction contained: 10 μL of IQSYBR Green Supermix (Bio-RAD), 0.8 μL of forward and reverse primers (5 μM each) of each target gene, 2 μL cDNA and 7.2 μL of water. The real-time program included an initial denaturation period of 1.5 min at 95 °C, 40 cycles at 95 °C for 15 s, and 60 °C for 30 s, followed by a melting step with ramping from 55 °C to 95 °C at a rate of 0.5 °C/10 s.

The specificity of the amplified PCR products was confirmed by agarose gel electrophoresis and melting curve analysis.

Gene relative expression was normalized according to the geometric mean of the reference genes. mRNA relative expression was analysed as fold increase using the 2^−ΔΔCT^ method [35], control group was constituted by VW-MSCs cultured in flask under standard 2d culture condition.

### 2.7. DAPI Staining and Immunofluorescence

To test the cellular distribution on the different scaffolds, VW-MSCs cultured for 72 h were processed for labelling DNA in fluorescence with DAPI staining and alpha-tubulin immunofluorescence. Biomaterials were washed twice in DPBS then were fixed overnight in cold 4% formaldehyde solution in PBS, pH 7.4. Each sample was transferred into a 25% sucrose (Sigma-Aldrich, MO, USA) solution in PBS at 4 °C for 24 h to get cryoprotection. Finally, samples were embedded and freezed in OCT (Sakura Finetek, CA, USA). Ten micrometres sections were cut at a Leica CM1950 cryostat (Leica, Wetzlar, Germany) mounted on microscope’s slides and stained with DAPI Staining Solution (Thermo Fischer Scientific, Rockford, IL, USA). For immunofluorescence staining slides were completely dried at room temperature, washed three times in PBS 1X for 5 min, permeabilized with Triton X-100 0.1% in PBS 1X for 1 h then washed three times in PBS 1X for 5 min. For aspecific sites blocking slides were treated with 10% Normal Goat Serum in PBS 1X for 1 h at RT then incubated overnight in a humidified chamber with an anti-alpha-tubulin antibody (Clone TU-01, Thermo Fisher Scientific, Waltham, MA, USA) diluted 1:250 in PBS 1X. In negative controls, the primary antibody was omitted. Then samples were washed three times in PBS 1X, incubated with anti-mouse FITC conjugate antibody (Sigma-Aldrich, St. Louis, MO, USA) diluted 1:100 in PBS 1X for 1 h at RT in a humidified chamber. After two washes for 5 min in PBS 1X and one wash in distilled water for 5 min, coverslips were mounted on slides with Fluoreshield with PI (Sigma-Aldrich, St. Louis, MO, USA). Photomicrographs were obtained using a Nikon digital camera installed on a Nikon (Nikon Inc., Melville, NY, USA).

### 2.8. Statistical Analysis

Three primary cell cultures derived from three different animals were used. Data represent the mean ± SD (or ± range of expression for qRT-PCR) of the three biological replicates. Data analysis was carried out with Prism 5.0 software (GraphPad Software Inc., San Diego, CA, USA). One-way analysis of variance (ANOVA) followed by Tukey post hoc comparison Test was used. *p* values < 0.05 were considered statistically significant.

## 3. Results

### 3.1. Surface Porosity Evaluated by the Morphometric Analysis on ESEM Images

The mean value of surface porosity valuated at 500× and 1000× was 45.49% and 51.08 for PLA scaffolds (Figure 1a) and 31.51% and 26.94% for PLA-10CaSi-10DCPD scaffolds (Figure 1b).

Surface porosity of PCL scaffolds estimated at 500× and 1000× magnifications was slightly higher when compared to PLA, namely 51.58% and 52.22%, respectively (Figure 2a).

Similarly, the porosity of scaffolds PCL-10CaSi-10DCPD was 41.42% and 41.65%, respectively (Figure 2b).

### 3.2. Cytocompatibility

To investigate the VW-MSCs ability to colonize biomaterials, cell seeding efficiency (CSE) was quantified (Figure 3). Results indicated a significant efficiency rate at 24 h, both for PLA and PCL matrix (≥90%), and the presence of additives did not influence the cell seeding efficiency.

### 3.3. Metabolic Activity

In order to measure the metabolic activity of VW-MSCs grew on the scaffolds, a MTT based assay method was performed on cells 24, 48, and 72 h post seeding. The results showed that VW-MSCs maintain a metabolically active state when seeded on all the materials. After 48 h cells seeded on PLA and PLA-10CaSi-10DCPD showed higher metabolic activity compared to cells seeded on PCL or PCL-10CaSi-10DCPD (Figure 4).

### 3.4. Effect of Biomaterials on VW-MSCs Markers

VW-MSCs cultured in PGM on the surface of different scaffolds expressed all the studied genes (Figure 5). Interestingly, qPCR analysis revealed a statistically significant downregulation of α-SMA in VW-MSCs cultured on PCL scaffolds, in relation to the VW-MSCs cultured under standard 2D condition (CTR) (Figure 5). The presence of CaSi and DCPD additives restored the basal level.

On the contrary, PDGFR-β always decreased in both PLA and PCL scaffolds in the presence of CaSi and DCPD.

### 3.5. VW-MSCs Distribution on Different Scaffolds

DAPI staining and immunofluorescence assay for alpha-tubulin confirmed cellular presence on all the experimental 3D scaffolds. Interestingly, a difference in cellular distribution was observed. On PLA-10CaSi-10DCPD scaffolds VW-MSCs grew individually (Figure 6c,e), whilst on PCL-10CaSi-DCPD scaffolds, VW-MSCs grew in spherical cluster (Figure 6d,e). The same distribution was also observed also for VW-MSCs cultured on the non-doped scaffolds (data not shown).

## 4. Discussion

Mineral doped scaffolds, composed of a biodegradable synthetic polymer and bioactive mineral fillers showed attractive properties for their use in bone tissue engineering [13,30]. Biodegradable synthetic polymers such as those based on poly-α-hydroxyl acids (as PLA and PCL) are currently used for scaffold design, as they are biocompatible, biodegradable, with an appropriate shelf-life, may be produced with a low cost process and may be adapted to the surgical site [13,15].

Highly porous poly-lactic acid scaffolds doped with bioactive calcium silicates and calcium phosphates have been recently designed [28,29] and demonstrated the ability to be colonized by oral derived mesenchymal stem cells and to stimulate their shift through osteogenic lineage [27,36]. Calcium silicates/poly-e-caprolactone 3D printed scaffolds designed for dental pulp tissues revascularization purposes showed good dental pulp stem cell proliferation and osteogenic differentiation [37].

As vascularization is a crucial factor when approaching any engineered tissue, new scaffolds, able to promote local angiogenesis, could represent a multi-approach strategy to obtain the best result in engineered tissue.

Designed scaffolds showed a uniform surface porosity. Both PCL scaffolds formulations (pure PCL and mineral doped polymer) revealed higher surface porosity of scaffolds than PLA formulation. These values were in line with two recently published papers [29,30].

Recent studies demonstrated that PLA and PCL formulations doped with CaSi were able to leach biologically relevant ions (Ca and OH) and to nucleate apatite on its surface [29,30]. Both ions are biologically relevant for new bone tissue formation, as Ca ions act as a powerful signal for mineralizing cells [38] and the alkaline environment demonstrated to increase apatite nucleation [27]. Moreover, Si is an important element for angiogenesis induction during the early phases of bone regeneration and new bone tissue formation [5,19]. Indeed, Si may be localized in young bone active calcification sites [39]. Si from CaSi (such as bioglasses, ackermanite, wollanstonite) is able to increase gene expression of VEGF and FGF, two important pro-angiogenic cytokines [20,37] and upregulate the downstream signalling of nitric oxide synthesis, genes and activity [19,40,41]. Several CaSi (ackermanite and wollanstonite) were able to trigger the secretion of angiogenic growth factors from fibroblasts, with the result of a higher number of blood vessel infiltration intro tissue-engineered scaffolds [41].

In the present study, VW-MSCs have been cultured on experimental 3D porous scaffolds based on PLA and PCL doped with high amounts (20% w/w) of reactive biointeractive mineral fillers (CaSi and DCPD). The used vascular stem cells derived from the thoracic aorta [10] represent an excellent model to test the angiogenic/anti angiogenic potential of different substances/materials [42].

The results of this paper clearly demonstrate that both experimental PLA- and PCL-based mineral doped scaffolds support VW-MSCs growth and colonization; VW-MSCs grown on all the scaffolds were metabolically active throughout the experimental time (72 h). Interestingly, at 48 h, VW-MSCs cultured on PLA-based scaffold showed an increase of metabolic activity respect to PCL-based scaffold regardless of the presence of the doping agents. The difference in metabolic profile found for PLA based scaffolds must be further investigated, considering that a different metabolic activity could represent a stem cells differentiation fate towards different specific lineages [43].

We previously demonstrated [11] that VW-MSCs cultured in 2D are characterized by a typical gene expression profile including mesenchymal stem cell markers, such as CD90 and pericyte markers such as alpha smooth muscle actin (α-SMA) [44], platelet-derived growth factor receptor-β (PDGFR-β) [45,46] and neuron-glial antigen 2 (NG2) [46]. Similarly, VW-MSCs cultured on both the experimental 3D scaffolds express these angiogenesis markers suggesting the possibility of a successful long-lasting colonization.

In the present study the cells isolated from the media of thoracic aorta and grown on the scaffolds showed CD90+. CD90 is expressed by the stromal lineage cells as pericytes and resident adventitial adult human mesenchymal stem cells [47]. In fact, the vascular wall is a source of multipotent and lineage-restricted progenitor cells, which are isolated from all the layers of large vessel, including aorta [9]. The functional capabilities of various stromal vascular fraction constituents during the process of neovascularization have been reported [48].

PLA and PCL scaffolds doped with CaSi and DCPD mineral fillers showed the expression of alpha-SMA whilst PCL scaffold (control) caused a reduction of alpha-SMA expression. The expression of α-SMA—the actin isoform typical of vascular smooth muscle cells [44]—indicates the angiogenic differentiation on the experimental scaffolds.

In contrast, PDGFR-β was significantly reduced in scaffolds enriched with reactive minerals CaSi and DCPD, mainly in the PCL mineral doped scaffold. On the contrary, PLA increased the PDGFR-β expression.

PDGFR-β is a particularly critical factor that regulates the recruitment and proliferation of vascular cells [45] and regulates angiogenesis by inducing the transcription and secretion of VEGF. PDGF is released mainly by endothelial cells and binds to the PDGFR-b on perivascular cells. PDGFR-β activation stimulates the proliferation of perivascular cells, resulting a powerful chemoattractant inducing cells to migrate and expanding the endothelial cell populations [49]. PDGF-β/PDGFR-β receptor role in angiogenesis has been well-described and correlated with the increase of vascularity and the vascular wall maturation. PDGFR- β is expressed by vascular smooth muscle cells and pericytes and plays an important role in these cells proliferation during vascular growth [50]. PDGFR-β expression from pericytes is necessary for their recruitment and integration in the vascular wall [51], the increase of the expression of this receptor is accompanied by enhanced vascular development, followed by angiogenic sprouting with abundant pericyte coating and increased expression of VEGF and receptor activation in endothelial cells [52].

The results of this study are particularly attractive, and further investigation, increasing the time of culturing could explain if the different levels of α-SMA and PDGFR-β expression could be associated with different spontaneous differentiation. Interestingly, the expression of NG2, early marker of pericytes [53,54,55], is significantly higher in all the scaffolds than in the plastic control indicating a stimulatory effect of the scaffolds towards an early stage vascularization.

Pericytes form an outer sheet around the endothelium and may contribute to the creation of microvessels on the surface and into the deep portion of scaffold after their placement in body sites, such as maxilla bone defect or peri implant and root bone pockets.

The distribution of VW-MSCs cultured on the experimental 3D scaffolds showed an individual cell growth on PLA-10CaSi-10DCPD whilst on PCL-10CaSi-10DCPD cells grown as compact clusters, which resemble the spheroids that could be obtained by hanging drop culture of VW-MSC. The difference in gene expression could be related to this different cell distribution. It is well known that MSCs phenotype is dramatically changed when culture condition shifts from 2D to 3D and also depends on the modality of the cell colonization and aggregation (development of multi-cellular aggregates, commonly known as spheroids) on the surface; while the difference in α-SMA and PDGR-β expression seems to be related to the presence of mineral additives.

Overall, we demonstrated the possibility of culturing VW-MSCs on 3D biodegradable highly porous mineral doped poly-α-hydroxy acids scaffolds. VW-MSCs spontaneously colonize these biomaterials and maintain a general undifferentiated phenotype in the first 72 h of culture with slight differences with control 2D culture. The experimental 3D scaffolds allow the expression of angiogenic markers by VW-MSCs after 72 h of culture. Further studies, with longer growth kinetics, are needed to investigate the differentiated fate of these cells grown on the experimental 3D culture systems containing CaSi and calcium phosphate for pulp and bone regeneration [56].

Being VW-MSCs an excellent in vitro model to study vascular remodeling, due to their strong angiogenic attitude, and the possibility of culturing VW-MSCs on these biodegradable scaffolds represents a strategy for biomedical tissue engineering.

In conclusion, the property of the experimental materials to be colonized by vascular stem cells and to allow the expression of angiogenic markers demonstrated the potential of the scaffolds to allow/induce the formation of new microvessels. The study supports their potential application in tissue regeneration as in oral bone defects and in pulp tissue regeneration, both conditions which require the fast formation of a new 3D vascular network.

## Figures and Tables

**Figure 1 nanomaterials-10-00243-f001:**
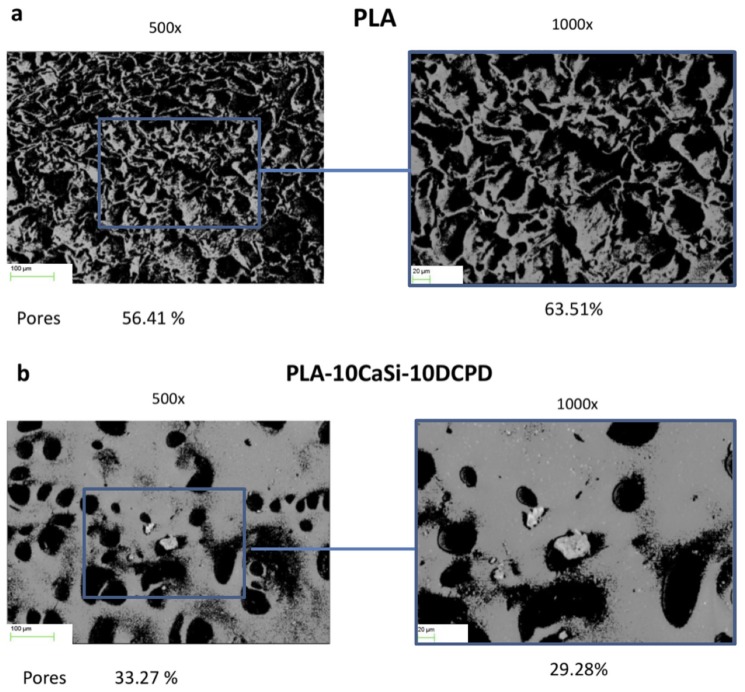
Surface porosity evaluation of PLA (**a**) and PLA-10CaSi-10DCPD (**b**) scaffolds on one random area at 500× and 1000× magnifications. Mineral doped formulation revealed a more compact surface, with a lower number of porosities when compared to pure PLA formulation.

**Figure 2 nanomaterials-10-00243-f002:**
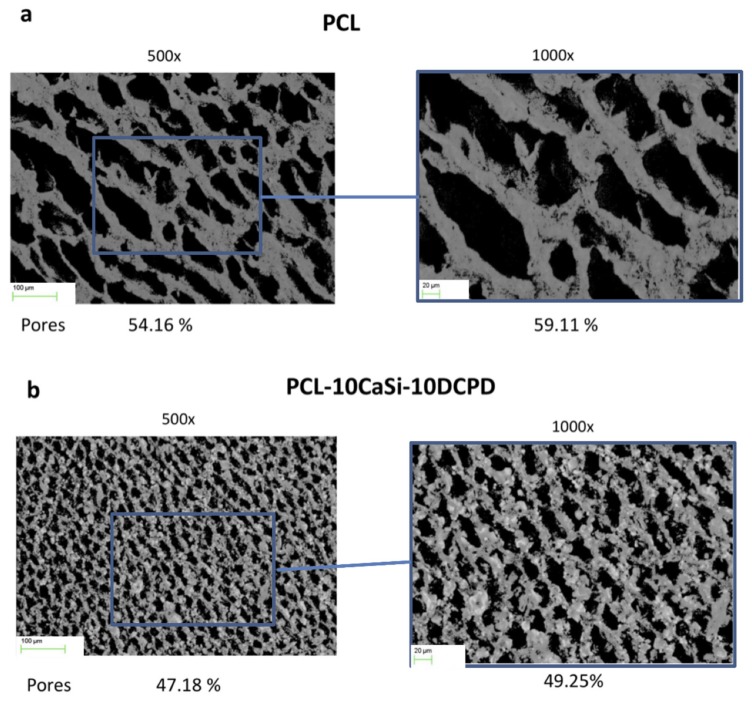
Surface porosity evaluation of PCL (**a**) and PCL-10CaSi-10DCPD (**b**) scaffolds on one random area at 500× and 1000× magnifications. PCL scaffolds reveal a more regular surface with larger porosities when compared to PLA scaffolds. CaSi and DCPD mineral fillers are widely distributed on the scaffold surface, partially occluding the pores.

**Figure 3 nanomaterials-10-00243-f003:**
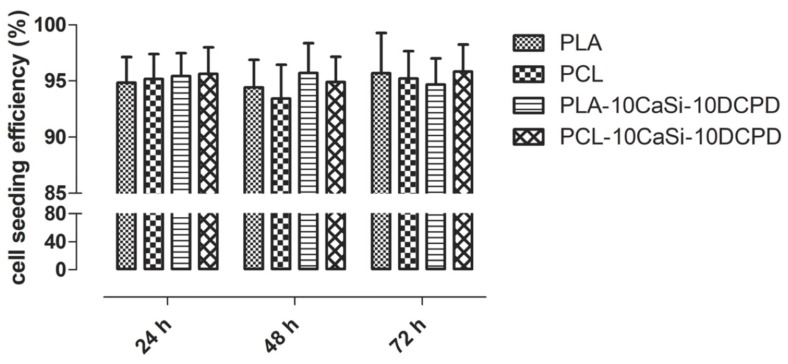
Cell seeding efficiency performed through an indirect method. After 24 h from the seeding on the scaffold surface, the unattached cells were quantified and cell seeding efficiency was calculated by the equation: CSE (%) = (1 − cells_u_/cells_i_) × 100. Data represent the mean ± SD of three independent biological replicates (n = 3) and were analysed using one-way ANOVA followed by the Tukey’s post hoc comparison. No differences were detected among different scaffolds.

**Figure 4 nanomaterials-10-00243-f004:**
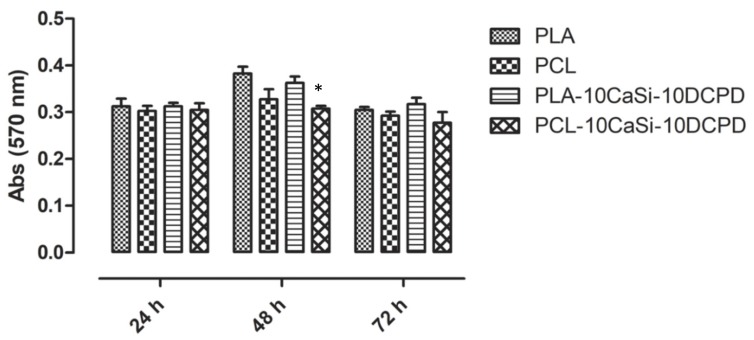
Metabolic activity of VW-MSCs seeded on the different scaffolds after 24, 48 and 72 h of culture was evaluated by MTT based assay. The data represented as the mean ± SD of three independent biological replicates (n = 3), were analysed using one-way ANOVA followed by the Tukey’s post hoc comparison. Significant differences among the experimental scaffolds are represented by asterisks (*****) (*p* < 0.05).

**Figure 5 nanomaterials-10-00243-f005:**
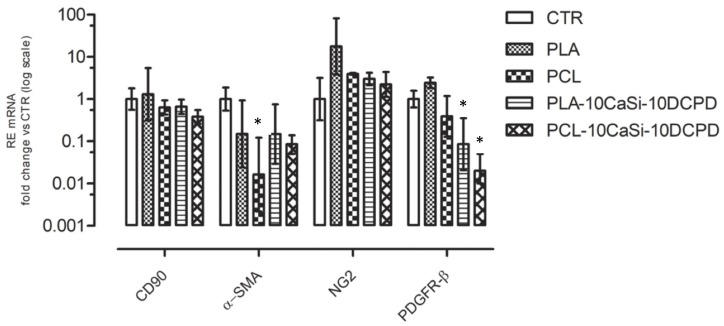
Expression of CD90, αSMA, NG2, and PDGFR-β genes in VW-MSCs evaluated after 72 h of culture in the presence of different experimental 3D scaffolds or in 2D standard culture condition (CTR). Data represent mean ± range of relative expression (RE) of three biological replicates (n = 3). Data were analysed using one-way ANOVA followed by the Tukey’s post hoc comparison test. Statistically significant differences among the scaffolds are represented by asterisks (*****) (*p* < 0.05).

**Figure 6 nanomaterials-10-00243-f006:**
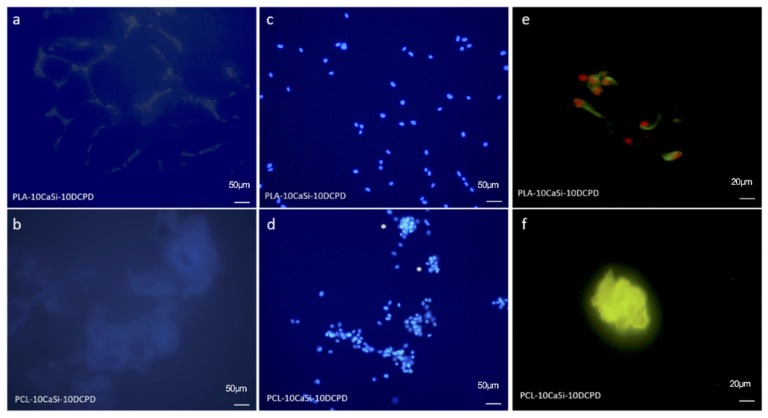
DAPI staining for PLA-10CaSi-10DCPD (**a**) and PCL-10CaSi-10DCPD (**b**) without cells and in the presence of 72 h cultured cells: PLA-10CaSi-10DCPD (**c**), PCL-10CaSi-10DCPD (**d**). Immunofluorescence analysis for alpha-tubulin of 72 h cultured cells on PLA-10CaSi-10DCPD (**e**) and PCL-10CaSi-10DCPD (**f**). Scale bar = 50 µm for a–d and 20 µm for e,f, *indicate spherical cluster.

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
