# Peer review of "Vascular Wall–Mesenchymal Stem Cells Differentiation on 3D Biodegradable Highly Porous CaSi-DCPD Doped Poly (α-hydroxy) Acids Scaffolds for Bone Regeneration"

_nanomaterials, 2020, doi:10.3390/nano10020243_

Round 1

Reviewer 1 Report

This work of angiogenesis in the field of Nanomedicine is well studied and discussed as well.

What is the main question addressed by the research? There are several mesenchymal stem cells which we could buy or get.

Did authors compared them for this study? Is it relevant and interesting? This paper may recognized a standard level of the recent tissue reengineering.

How original is the topic? Some gene expression in 3D culture

What does it add to the subject area compared with other published material? Some genes was well expressed in 3D culture

Is the paper well written? It may be a moderate levels of the field of nanomedicine

Is the text clear and easy to read? easy

Are the conclusions consistent with the evidence and arguments presented? Almost consistent

Do they address the main question posed? Almost reasonable

Author Response

Referee 1

This work of angiogenesis in the field of Nanomedicine is well studied and discussed as well.

What is the main question addressed by the research? There are several mesenchymal stem cells which we could buy or get.

Did authors compared them for this study? Is it relevant and interesting? This paper may recognized a standard level of the recent tissue reengineering.

How original is the topic? Some gene expression in 3D culture

What does it add to the subject area compared with other published material? Some genes was well expressed in 3D culture

Is the paper well written? It may be a moderate level of the field of nanomedicine

Is the text clear and easy to read? easy

Are the conclusions consistent with the evidence and arguments presented? Almost consistent

Do they address the main question posed? Almost reasonable

We thank the referee for the comments, manuscript has been revised in order to be more clear.

Reviewer 2 Report

In this article, Forni and collaborators present different biomaterials for embedding Mesenchymal Stem Cells. The question raised by the authors is interesting but the article does not bring good quality results.

Major comments :

all figures need to be reshaped, there are no scales, no magnified areas for helping readers to understand the results. immunostainings are of very poor quality english writing needs to be corrected

Author Response

Referee 2:

The question raised by the authors is interesting but the article does not bring good quality results.

Major comments :

all figures need to be reshaped, there are no scales, no magnified areas for helping readers to understand the results. immunostainings are of very poor quality English writing needs to be corrected 

The figures have been modified in order to be more clear. In addition, scale bars have been added In Figures 1,2 and 6, as well as magnified areas in Figures 1,2. High resolution (600dpi) figures have been provided for Figure 6. The English writing has been revised according with the referee suggestions.

Round 2

Reviewer 2 Report

Thank you for the corrections but I still consider that all images from Figure 6 have to be changed.

Author Response

To the Editorial board of Nanomaterials,

following the first reviewer indication, we have improved the quality of images to 600dpi. The new reviewer request appears to relate more to the content of the figure. We agree with the reviewer that the definition of the immunochemistry is quite low but, unfortunately, a confocal microscope was not available.

Considering that the principal intent of figure 6 is to explain the different distribution of cells in the two biomaterials, and not to describe the markers expression (which we have done by real time PCR), we believe that the relative images (Fig 6e and f) are in any case explanatory. If you do not agree, we are eventually willing to remove the immunocytochemistry panel. Finally, to avoid any misunderstanding in the interpretation, we have also deleted the following comment in the text:

“with interconnection by cytoplasmic prolongations” (line 338):

Response to the reviewer

We are thank to the reviewer for the suggestion; however, the only function of figure 6 is to show the presence of live cells 72h after seeding and to provide an indication of their distribution in the biomaterials. Therefore, we have modified the plate (fig 6c-d) to show a larger field in which the different distribution of cells grown on PLA-10CaSi-10DCPD respect to cells grown on PCL-10CaSi-DCPD is well evident.

We have re-written the figure caption

Figure 6. DAPI staining for PLA-10CaSi-10DCPD (a) and PCL-10CaSi-10DCPD (b) without cells and in the presence of 72 h cultured cells: PLA-10CaSi-10DCPD (c), PCL-10CaSi-10DCPD (d). Immunofluorescence analysis for alpha-tubulin of 72 h cultured cells on PLA-10CaSi-10DCPD (e) and PCL-10CaSi-10DCPD (f). Scale bar = 50 µm for a-d and 20 µm for e-f, *indicate spherical cluster.

Finally, to avoid any misunderstanding in the interpretation of immunohistochemistry, we have deleted the following comment in the text

“with interconnection by cytoplasmic prolongations” (line 338):

Round 3

Reviewer 2 Report

Dear authors 

thank you for your efforts. 

Author Response

We thank  the referees for all the provided suggestions